# Wearable Activity Trackers in the Management of Rheumatic Diseases: Where Are We in 2020?

**DOI:** 10.3390/s20174797

**Published:** 2020-08-25

**Authors:** Thomas Davergne, Antsa Rakotozafiarison, Hervé Servy, Laure Gossec

**Affiliations:** 1Sorbonne Université, INSERM, Institut Pierre Louis d’Epidémiologie et de Santé Publique (UMRS 1136), 75013 Paris, France; laure.gossec@aphp.fr; 2APHP, Rheumatology Department, Pitié Salpêtrière Hospital, 75013 Paris, France; antsaniriantsoa.rakotoalson@aphp.fr; 3E-Health Services Sanoïa, 13420 Gémenos, France; hservy@sanoia.com

**Keywords:** wearable activity trackers, disease flares, remote monitoring, rheumatic disease

## Abstract

In healthcare, physical activity can be monitored in two ways: self-monitoring by the patient himself or external monitoring by health professionals. Regarding self-monitoring, wearable activity trackers allow automated passive data collection that educate and motivate patients. Wearing an activity tracker can improve walking time by around 1500 steps per day. However, there are concerns about measurement accuracy (e.g., lack of a common validation protocol or measurement discrepancies between different devices). For external monitoring, many innovative electronic tools are currently used in rheumatology to help support physician time management, to reduce the burden on clinic time, and to prioritize patients who may need further attention. In inflammatory arthritis, such as rheumatoid arthritis, regular monitoring of patients to detect disease flares improves outcomes. In a pilot study applying machine learning to activity tracker steps, we showed that physical activity was strongly linked to disease flares and that patterns of physical activity could be used to predict flares with great accuracy, with a sensitivity and specificity above 95%. Thus, automatic monitoring of steps may lead to improved disease control through potential early identification of disease flares. However, activity trackers have some limitations when applied to rheumatic patients, such as tracker adherence, lack of clarity on long-term effectiveness, or the potential multiplicity of trackers.

## 1. Introduction

All areas of our lives are influenced by digitalization and connected devices, and this includes healthcare [1,2]. E-health encompasses traditional telemedicine—where a doctor performs a diagnosis or procedure remotely—but also patient information and self-monitoring, or continuous remote monitoring with connected devices or mobile applications [3]. E-health is also linked to new epidemiological and statistical possibilities based on analyzing massive and heterogeneous data (Big Data), using artificial intelligence [4,5]. The growth of e-health is due in part to the wide variety of health-related trackers available. In the field of rheumatology, currently, most developments of connected devices are centered on physical activity [6]. In health management, physical activity monitoring could be used in two ways: for self-monitoring by the patient himself or for external monitoring by health professionals. There is an increasing popularity of activity trackers amongst people with long-term conditions, because the role of the patient as an actor in his/her own care is growing, in a context of increased shared decision-making [7,8]. Furthermore, activity trackers can support disease assessment by health professionals and potentially enhance doctor–patient interactions [1]. Thus, activity trackers can have various uses in patients with musculoskeletal and rheumatic diseases.

There are two main types of rheumatic and musculoskeletal diseases: mechanical ones (such as osteoarthritis or low back pain), and inflammatory ones (including rheumatoid arthritis and spondyloarthritis) [9]. Although osteoarthritis or low back pain are more frequent, there are more treatment options and a more protocolized management for inflammatory diseases. In this paper, we mainly report the use of activity trackers for inflammatory diseases.

Inflammatory arthritis is the consequence of an autoimmune disease where joints or the spine become inflammatory. Rheumatoid arthritis and spondyloarthritis have several common features (Table 1), though they differ in their clinical presentation. They also both lead to important impact on patients’ lives with chronic pain, functional disability (including difficulties walking or performing daily activities), and fatigue [10]. The prevalence of these two conditions is 0.5% and 0.3%, respectively.

One of the key aspects of management of these conditions is adherence to healthy behaviors, such as regular sleep, sufficient physical activity, or stress reduction [11,12,13,14]. Activity trackers have the potential to track these behaviors in order to motivate and educate patients.

Activity trackers can also be used in patients with musculoskeletal and rheumatic diseases as a tool for external monitoring. Many innovative electronic tools are currently used in rheumatology to support physician time management, to reduce the burden on clinic time, and to prioritize patients who may need further attention [15]. Electronic self-follow-up appears to be feasible and acceptable for patients in rheumatology [16]. Connected devices can allow automated self-monitoring without patient intervention especially when the data collection is passive [17]. In inflammatory arthritis, flares are fluctuations of disease activity levels, which have consequences on long-term outcomes. In these conditions, regular or even continuous monitoring of patients to detect disease flares is one of the keys of a treat-to-target approach which is recommended by international consensus guidelines [18,19,20]. However, regular assessments in clinics are time-consuming and complex. E-health can allow continuous remote monitoring using connected devices and mobile apps [21]. Such electronic assessments could facilitate prioritization of patients to receive rapid medical attention, and provide a more detailed record of the disease course over time between visits.

### Objectives

In this review, we discuss the roles of connected activity trackers for the management of patients with rheumatic and musculoskeletal disorders, in 2020. First, we discuss the use of activity trackers for self-monitoring to promote physical activity in patients at risk of inactivity. The need to consider behavior change techniques and address barriers and facilitators, in addition to the use of activity trackers, is explained. We then discuss the use of activity trackers as a promising tool for detecting flares in patients with inflammatory arthritis. We introduce the use of machine learning to enable more accurate analysis of data to detect flares. Although the promising use of activity tracker for self-monitoring and external monitoring, various limits exists, and are addressed at the end of this review.

## 2. Use of Activity Trackers for Self-Monitoring of Physical Activity

The number of small, relatively inexpensive mobile devices is growing [22]. These devices allow an individual to track many measures of daily life. This area is often referred to as “quantified self” and many of these measures have potential implications for healthcare management and may influence behavior to promote a healthier lifestyle. Example of measure which can be tracked include physical activity, sleep patterns, UV exposure, or heart rate [23]. This section will address the use of activity trackers for self-monitoring of physical activity since this subject is the most developed in rheumatology [24,25].

### 2.1. Physical Activity: Definition and Recommendations

Physical activity is usually defined as “any bodily movement produced by skeletal muscles that results in energy expenditure” [26]. Physical inactivity and sedentary behavior are two different notions with the first one defined as “an insufficient physical activity level to meet present physical activity recommendations” [27] and the second one as “any waking behavior characterized by an energy expenditure ≤1.5 metabolic equivalents (METs), while in a sitting, reclining or lying posture” [27]. It is important to note that a person can be physically active by doing 30 min of moderate physical activity in the morning, but at the same time sedentary by remaining seated for the rest of the day [26].

The World Health Organization’s recommendations on physical activity and sedentary lifestyles for adults are outlined in Box 1 [28]. Thus, 150 min of physical activity are, per week, recommended. This is often approximated as 10,000 steps per day [21].

Box 1Recommendations according to the World Health Organization for adults between 18 and 64 years.For physical activity: equivalent to 150 min of moderate-intensity endurance activity or at least 75 min of sustained-intensity endurance activity (performed in periods of at least 10 min) during the week. For additional health benefits, adults should increase the duration of moderate-intensity endurance activity to 300 min per week or 150 min per week of sustained endurance activity, or an equivalent combination of moderate and sustained activity. Strengthening exercises involving the major muscle groups should be performed at least two days per week.For sedentary lifestyles, it is recommended: to reduce the total daily time spent sitting or lying down (outside of sleeping and eating time) as much as possible. To walk for a few minutes and stretch after 2 h in a row in a sitting or lying position and to make a few movements that activate the muscles and mobilize the joints (rotation of the shoulders, pelvis, ankles, wrists, hands, head).

### 2.2. Why Is Physical Activity Important in the General Population and in Rheumatology Patients?

Active living is recommended at any age. Adopting an active lifestyle means increasing physical activity and reducing sedentary time.

Adopting an active lifestyle protects against many non-communicable diseases and reduces mortality (Box 2). In patients suffering from osteoarthritis or chronic inflammatory rheumatic diseases, general exercise, aerobic activity, strength exercises, or yoga sessions are linked to a significant reduction in pain, depression, disease activity, and improvement in cardiovascular disorders, joint mobility, and physical function [29,30,31,32].

Box 2Overview of benefits of physical activity in the general population [33].Decrease in coronary heart disease: 6%Decrease in type II diabetes: 9%Reduction of breast cancer: 10%Reduction of colon cancer: 10%Decrease in premature deaths: 9%

### 2.3. Lack of Physical Activity in the General Population and Patients with Musculoskeletal and Rheumatic Diseases

Despite the benefits of physical activity, healthy people and patients are vulnerable in terms of physical activity levels. According to the World Health Organization in 2016, 23% of men and 32% of women aged over 18 were insufficiently physically active [34]. Over the past 15 years, levels of insufficient activity did not improve. This assessment is even lower for patients with inflammatory rheumatic diseases or osteoarthritis, with, for example, 1 in 4 adults reaching the recommendations for patients with spondyloarthritis in the United Kingdom [35]. When physical activity is objectively measured using an activity monitoring system, people with chronic musculoskeletal conditions are approximately 60 min less active per week than healthy adults, as shown in an observational study of 96,706 participants with and without chronic conditions [36]. For rheumatoid arthritis, this difference is even bigger with a difference of 200 min compared with healthy persons (Figure 1).

### 2.4. Wearable Activity Trackers to Promote Physical Activity

Wearable activity trackers, also known as consumer electronic devices and connected devices, are devices capable of tracking a person’s physical activity. The first generation of trackers was represented by pedometers and could only track the number of steps [37,38]. For the last 10 years, the new generation of activity trackers has also made it possible to track the number of steps taken, the time spent inactive, to dissociate intense activity from moderate activity and to calculate energy expenditure [39]. The connection with other devices such as a smartphone or computer also allows this information to be displayed in the form of graphs and informative statistics [40]. Such features offer a remarkable possibility for external monitoring or self-monitoring by passively, automatically and continuously collecting physical activity. Wearable activity trackers are substantially growing in popularity. The activity trackers market size was valued at $17,907 million in 2016, and is expected to reach $62,128 million by 2023 [41].

Wearable activity trackers are emerging solutions to improve physical activity levels and decrease inactivity. Via sensors, these devices help users track their daily movement and provide feedback on activity with monitor displays, companion smartphone tools, computer applications or websites associated with the device [42]. This technology aims to educate and motivate users toward better physical activity habits and better health behavior [43].

Improving physical activity behavior requires behavior change techniques (BCTs). Interventions involving wearable activity trackers appear to be linked to specific BCTs. Indeed, a study analyzing 13 wearable activity trackers identified an average of nine techniques, with the most common techniques implemented being self-monitoring, providing feedback, adding objects to the environment, and goal setting [44]. Another study analyzing seven wearable activity trackers identified a mean of 16 BCTs, with 9 techniques present in every tracker [42]. Moreover, interventions involving wearable activity trackers are typically complex, with multiple components, usually associated with a surrounding delivery package [44,45].

Self-monitoring of physical activity is important to understand and become aware of the level of activity or the length of time sedentary per day. This amount of activity or sedentary time can easily be underestimated [46,47]. When individuals are asked to quantify their level of physical activity, they generally tend to overestimate this level compared to an objective measure, indicating that it is difficult to be aware of one’s activity level [46]. Self-monitoring appears to be one of the most widely used and effective techniques for changing behavior and is included in several recommendations for changing a behavior that is important for health, namely physical activity [42,48].

### 2.5. Accuracy of Activity Trackers

Although wearable activity trackers have seen an upward trend worldwide, there are concerns over the accuracy of the data collected by these devices. Another issue with accuracy of activity trackers is that there is no consensus on standard procedures to validate accuracy of activity tracker [6]. However, one study including 104 participants conducted an experience in free-living conditions during 7 days. The average number of steps per day was compared between the wrist-worn Fitbit Flex and waist-worn ActiGraph (wGT3X-BT) considered here as a reference standard [8]. This study showed a high correlation and agreement in steps between Fitbit Flex and ActiGraph, while discrepancies were observed between devices in average steps/day (Fitbit Flex, 10193; ActiGraph, 8812).

### 2.6. A Systematic Review Assessing Activity Trackers to Increase Physical Activity in Rheumatic Patients

A systematic review of interventions to increase regular physical activity with a wearable activity tracker was conducted in patients with rheumatic and musculoskeletal diseases (17 studies; 1588 patients) [49]. Overall, 9 studies used a pedometer (KenzLifecoder EX, Suzuken Co., Yamax Digi-Walker Pedometer Model SW-200 (x2), Yamax Digi-Walker CW-701, Omron HJ720ITC (x2), Mt-x movement sensor and a PDA or non-specified) and 8 a more advanced activity tracker (Jawbone UP 24, Fitbit (Fitbit Flex, Fitbit Charge HR, Fitbit Zip (x2), or non-specified (x3)). Activity trackers were worn on the wrist or waist. The results showed that short-term adherence to wearable activity trackers was high in this population. Here, adherence was considered as (duration of use and percentage of patients still in the study at completion [study completion rate]). Adherence was 92% at 10 weeks. Regarding the effect of the activity tracker on increasing physical activity levels, results showed an interesting increase of 1520 steps per day between groups using the tracker and groups without the tracker. These results are based on a meta-analysis of seven studies (463 patients) after an average wearing time of 14 weeks. An effect was also observed on the time spent in moderate to vigorous physical activity with a 16 min daily difference between groups. However, no significant results were found for the main outcomes in prolonged follow-up after the end of the intervention period. Increase in physical activity was not correlated with an increase in short-term symptoms, although pain increased during long-term interventions.

Other systematic reviews of the literature showed promising effect of activity trackers to increase physical activity in various populations such as the general population [50,51], young populations [52] and adults who are overweight or obese [53].

### 2.7. Barriers and Facilitators to Physical Activity Should Be Addressed

Increasing physical activity is a challenge [54,55]. In addition to the use of new technologies to increase physical activity, other aspects should be taken into consideration such as barriers and facilitators, stages of behavior change for a global approach [56,57,58]. Regarding physical activity, barriers and facilitators have been identified [57,58]. Barriers appear to be mostly related to psychological status such as fear of movement, and facilitators are linked in part to social support such as receiving encouragement to participate in physical activity or having a partner to play sports with [57,59]. The concept of stages of behavior refers to the readiness of a person to engage in a specific behavior (precontemplation, contemplation, preparation, action and maintenance) [60]. It has been shown that interventions that aim to increase physical activity are more efficient if matched to the stage of behavior [61]. These elements are important to consider in a comprehensive approach to improve physical activity and should be associated with the use of activity trackers.

## 3. Activity Trackers as Tools to Monitor Disease Activity in Chronic Rheumatic Diseases

Regular monitoring is necessary in inflammatory rheumatic diseases, such as rheumatoid arthritis or spondyloarthritis [17,62,63]. Activity trackers may be useful for patient monitoring by physicians and, thus, may lead to optimization of disease control in inflammatory arthritis. In this part of the manuscript, we will review why and how activity trackers may be used to monitor disease flares.

### 3.1. Why Are Flares Important in Inflammatory Arthritis?

The evolution of chronic rheumatic diseases is marked by alternated periods of flares and stable disease activity [64]. Even in patients receiving optimal and costly treatments to regulate the immune system, flares are frequent [65,66]. Flares correspond to a worsening in disease activity. According to patient descriptions, the term “flare” can correspond to multiple situations such as increased symptoms within normal variation or unprovoked increased symptoms that are unmanageable, persistent, leading to seeking help. Therefore, a flare represents a cluster of symptoms of sufficient duration and intensity to require changes in therapy [65,67,68].

In periods of flares, patients present with worsened pain and symptoms but also higher inflammation, which may be assessed through physical examination (e.g., through an increase in joint swelling). The assessment of these flares is important in clinical practice to better understand disease status and treatment efficacy [69,70]. Fluctuations in disease activity have deleterious consequences in the short and long-term [71,72]. Temporary variations in disease activity are linked to radiographic progression (meaning an increase in joint or spine alteration, which is irreversible) [71,72,73]. The poor control of chronic inflammation in rheumatic diseases is associated to a worsening of symptoms and may lead to ulterior functional disability (e.g., reduction of functioning and ability to perform activities), which have an impact on patients’ quality of life (Figure 2) [69,72].

According to international guidelines, physicians must aim for remission or low disease activity (i.e., good control of inflammation) since this leads to improved outcomes in rheumatic diseases such as rheumatoid arthritis [19,62,63]. It is widely accepted that clinical remission is the main therapeutic target for patients with rheumatoid arthritis, with low disease activity as a best possible alternative. In axial spondyloarthritis, the objective of treatment is also remission though the definition of remission is still the object of discussions. In this context, flares must be monitored and controlled. The ‘treat to target’ strategy implies monitoring disease activity and adjusting treatment regularly to obtain full control of inflammation [19,62,63]. Flares are of interest to patients and clinicians since they reflect disease activity fluctuations, which should be avoided when applying the ‘treat-to-target’ approach. When a patient is in flare, the recommendations are to assess the patient within a few days and to adjust treatment rapidly. Such assessments should then be repeated regularly e.g., once a month, as long as disease control is imperfect [62,63].

### 3.2. How to Assess Flares?

Flares can be assessed by single questions or standardized questionnaires to detect increased inflammation or life changes. (Table 2) [74,75]. For this assessment, the physician may ask questions directly to the patient during the clinical visit, or the patient may complete a questionnaire himself or herself without the physician’s assistance.

There is no consensus on the best way to assess flares. Changes in disease activity status are the usual ways to detect flares, but necessitate a patient visit to the clinic with physical examination and blood tests, which is not always feasible within a very short time frame [67]. Questionnaires are also useful to assess flares, and can be filled in from home. Remote monitoring of questionnaires (called patient-reported outcomes) may facilitate a treat-to-target approach and help to measure flares, since it is possible with remote patient monitoring to perform regular assessment of disease status [17]. However, questionnaires completed remotely online necessitate patient engagement in care, which may lessen over time [77].

As explained above, physical activity and in particular walking, is an essential part of daily life and may be influenced by flares. Indeed, flares lead to physical pain and alteration for joint and spine function, which may lead to decreased physical activity but also modifications in activity patterns [78]. Physical activity can be objectively measured using activity trackers, although there is some uncertainty about the accuracy of measurement (see Section 2.5) [21]. Thus, we hypothesized that activity trackers could be used in the assessment of disease flares. We performed a study to test this hypothesis, which we will describe briefly below [21,65,68].

### 3.3. Detecting Flares by Activity Trackers: The ActConnect Study

We performed a 3 months longitudinal observational study named the ActConnect study in 2018. The objective of the study was to evaluate flares in rheumatoid arthritis and axial spondyloarthritis based on repeated weekly assessments, and to explore the link with physical activity measured by activity trackers [21,65,68].

Patients with either rheumatoid arthritis or axial spondyloarthritis were included as well as 20 healthy controls. A total of 157 patients (83 rheumatoid arthritis and 74 axial spondyloarthritis) were analyzed; 36.3% patients were males with mean age of 46 (standard deviation [SD] 12) years and mean disease duration of 11 (SD 9) years. Patient-reported flares were assessed weekly through the patient’s smartphone by asking a dedicated question: “Has your disease flared up since the last assessment?”, with a categorical response according to no flare, flare lasting 1–3 days (short flare), or flare lasting more than 3 days (persistent flare). Physical activity was collected continuously using a connected activity tracker (Withings^®^ Activity Pop watch) over the 3 months. The watch was provided for free to the participants and they were asked to wear it continuously for the duration of the study.

### 3.4. The Main Results of ActConnect

#### 3.4.1. Flares Were Frequent

Most of the 170 patients had long-standing disease and around half of them were receiving biologic therapy (strong and costly immunosuppressive drugs). Although the disease appeared well controlled, we found that flares were frequent: patients reported having experienced a flare on average in 28% of the weekly assessments. Short flares were more frequent than persistent flares, corresponding to 26 flares for 100 patient-weeks [65].

#### 3.4.2. Physical Activity Was Moderate

The mean number of steps per day over 3 months was 7124 (standard deviation: 2316) corresponding to 108 (36) minutes per day of moderate to vigorous activity (Figure 3). Thus, physical activity was moderate overall, with 24–30% of patients fulfilling the World Health Organization recommendations for physical activity [21,28].

#### 3.4.3. Link between Flares and Steps

In a first phase, the relationship between physical activity and disease activity was assessed using the *nlme* package in R with univariate and multivariate analyses and linear mixed-effect models. The *nlme* package (for Linear and Nonlinear Mixed Effects Models) allows to fit and compare Gaussian linear and nonlinear mixed-effects models [79]. We found that persistent flares were related to a moderate decrease in physical activity [65]. At the group level, there was a relative decrease in physical activity of 12–21% during weeks with flares, corresponding to an absolute decrease of 836–1462 steps per day [21,65]. However, using standard statistics, we were unable to find a precise cut-off value allowing to detect flares based on steps because it was difficult to quantify precisely this reduction of daily step and only indicative ranges could be presented, according to the possible flare duration.

#### 3.4.4. Use of Machine Learning to Enable More Accurate Analysis

In a second phase, we analyzed the link between patient-reported flares and activity-tracker-provided steps per minute (and not mean steps per day) using artificial intelligence, and more specifically machine-learning through selective (multiclass) naive Bayesian statistical methods [68]. Machine learning allows analyses of huge amounts of data with minimal aggregation of data. Such techniques necessitate high computer power and dedicated programs. In this case, we worked with a telecommunications company, Orange Healthcare.

The patient’s walking profile and the corresponding patient response to the flare question over the first weeks were used to ‘calibrate’ the machine, at the patient level. Then variations in steps and in their patterns were analyzed and (in very simple terms for non-statisticians) the machine told us if the patient was flaring or not [5,68,80]. This was compared to the patient response to the flare question, used here as gold standard. The modelling phase, an evaluation of the importance of each explanatory variable in the model was provided by the software of machine learning used (Khiops©). Using these evaluation, a time-line map of “significant” moments of activity during the week was created (data not shown) [68].

It is remarkable that the machine learning model detected correctly both patient-reported flares and absence of flares with a sensitivity (the ability of a test to correctly classify an individual as flaring) of 96% and a specificity (the ability of a test to correctly classify an individual as not flaring) of 97% [81]. The corresponding positive and negative predictive values were respectively 91% and 99% (Table 3).

#### 3.4.5. What Are the Practical Implications of Our Findings?

We believe there are three main take-home messages in this study.

Firstly, these results confirm objectively the reality of the impact of patient-reported flares, using an objective measure of daily life and functioning. This is important given that some healthcare professionals do not ‘believe’ patient-reported flares are true flares [64,69].

Secondly, the results indicate that wearable activity trackers may give indirect information on disease activity. Thus, activity trackers may be useful for more than measuring steps and motivating patients to move more [49]. We found patient-reported flares were strongly linked to physical activity and that patterns of physical activity could be used to predict flares with great accuracy. Automatic monitoring of steps may lead to improved disease control through potential early identification of disease flares, with high convenience for patients since the data collection is passive [82].

Potential practical applications in the clinic of these findings would include a three-step process. Firstly, the patient would wear an activity tracker linked to the healthcare center. In cases of decreased physical activity, an alert would appear for the healthcare professional. Then, the center would contact the patient to check if indeed he/she is flaring or if the decrease in physical activity is not due to a flare (e.g., holidays lying down on a beach!). Thirdly, if a flare was confirmed by the patient, a visit would be organized so that the patient could be assessed fully and treatment changed if necessary. Of course, this theoretical framework has caveats, including data confidentiality, anonymity and privacy concerns as well as responsibilities taken on by the healthcare professionals in relation to alerts (e.g., what happens if the physician does not contact a patient after an alert, and the patient was having a heart infarct?) [5,80,83].

The final key finding here is that this study is one of the first to demonstrate the usefulness of machine-learning applied to large rheumatology datasets [5,80].

We believe the next years will see an explosion in such analyses.

## 4. Limits of Activity Tracker in Clinical Practices

### 4.1. Limits of Activity Trackers to Self-Monitor Physical Activity

Adherence to activity trackers could be low, reducing the potential effect of the tracker. More than half of the participants stop using the activity tracker after two weeks and 75% after four weeks as showed in a study in undergraduate students [84]. However in rheumatic diseases, short term adherence to trackers was excellent [49]. The discrepancies of results between these two population could come from extra efforts to maintain adherence in randomized clinical trials (e.g., weekly phone calls) [85,86].

Long-term effect of activity trackers. The question of the effectiveness of wearable activity trackers over the long-term remains unsolved. In two studies with follow-up after end of intervention, no evidence of increase in steps after stopping wearing the wearable activity trackers was observed [87,88].

Activity trackers measure only a small proportion of total physical activity. Recent activity trackers can measure various forms of physical activity such as number of steps, stairs, and to identify energy expenditure [89]. However, a wide variety of physical activities are not covered by most activity trackers, such as swimming or arm movement when the tracker is worn at the waist [90]. A substantial part of the physical activity can be related to domestic activity such as gardening or cleaning. These activities are generally not well captured by the activity tracking system, leading to an unrepresentative measure of physical activity [91].

Health literacy and physical literacy may influence the interpretation of data collected by trackers. The concept of health literacy refers to the personal and relational factors that affect a person’s ability to acquire, understand, and use information about health and health services [92]. Similarly, physical literacy can be described as a disposition to take advantage of embodied human capacity in which the individual has the motivation, confidence, physical competence, knowledge, and understanding to value and take responsibility for maintaining meaningful physical activity throughout life [93]. It has been shown that active people have a higher health literacy than inactive people [94]. In addition to providing monitoring of activities to increase physical activity, a comprehensive approach could also focus on health and physical literacy. These skills could be developed through patient education [95].

Accuracy of activity trackers. As mentioned above in point 2.5, the accuracy of activity tracker remains a concern and must be taken into account when analyzing results. The accuracy of step counting can be influenced by the walking speed. In a study comparing the number of steps measured by commercial activity trackers versus gyroscopic readings at 2 km/h, the highest error showed a margin of 26.8% with the Fitbit HR load, while at 3.5 km/h, the highest error showed a margin of 1.5% with the Fitbit One [96].

### 4.2. How to Manage Trackers in a Busy Clinic?

The use of activity trackers in clinical practice where every day each physician treats up to 30 patients, requires an approach that would guarantee ease of use and standardization of data from their sampling to their annotation [97]. One way could be the Internet of Things (IoT) integration services, that would allow patients to use their own device vendors technical stacks and disparity of devices are challenges, despite notable initiatives for instance at the smartphone level [98]. The second possibility would be a single, standardized track, but it also presents a number of difficulties:−paradox of messaging: user acceptance and the psychological effect of a tracker: The objective for the medical community is to ensure remission of the disease and to make it as discreet as possible, with the least possible impact on quality of life. Wearing a dedicated medical device has an impact on the perception of the disease and can remind the patient of his or her condition [99];−healthcare agencies momentum: healthcare organizations need to develop an entirely new regulatory paradigm to design a framework for digital opportunities. Indeed, a standardized device would be produced after a long public process with numerous accreditations, and therefore with a delivery to patients of devices that are 2 or 3 years old [100];−trackers like a pill: today, most activity trackers rely on a smartphone as gateway. Beyond smartphone adoption, smartphone effective use is a challenge for part of the patients, and diversity of Bluetooth versions or core system versions across devices could be challenging. Meaning we should not underestimate the assistance required to help patient to pair the device to his smartphones or re-pair it if lost. Moreover, 5G (the fifth generation technology standard for cellular networks) low-power WAN, which is a type of wireless Wide Area Network designed to enable long-distance communications at low data rates, do not require any smartphone to operate. Using such networks would be a significant step in the generalized use of trackers in daily care allowing to prescribe trackers like pills [101].

### 4.3. The Patient in 2023: A 5 Wristband Owner?

With the progress in terms of early detection and care of chronical disease, a significant number of patients has more than one disease [102]. Auto-immune diseases such as inflammatory arthritis could have an impact on multiple organs, and also on mental health [103]. The integrated approach is not yet the norm in care and this requires an effort on the part of patients to coordinate their care. The risk of continuing with a “one device–one disease” logic and generating a new burden due to multiple devices to be carried must be considered [104].

#### 4.3.1. Inspire and Share

Not all medical fields are at the same stage when it comes to talking about monitoring activities or disease markers. Sharing experiences and examining progress in other medical fields could be a great source of inspiration. For example, in the management of diabetes, in ten years we have gone from continuous blood glucose measurement with a modern device to remote monitoring [105] reimbursed as an ordinary medical procedure [105,106]; and now, to innovative software that merges the data from a continuous glucose monitor, processes it with artificial intelligence to send the command to the subcutaneous insulin pump in a “closed-loop” system integrating “physical activity” measured with glucose consumption [107].

#### 4.3.2. Listen to Real-Life?

Should an activity tracker, as a device to wear, be the future? What about listening and processing data information that surrounds us? Analyzing existing data could be used as markers for specific medical outcomes (e.g., the human voice and coronary artery disease) [108]. How technical innovation could inspire medicine (i.e., detecting human moving with Wi-Fi) is an area to urgently consider [109].

## 5. Conclusions

Wearable activity trackers are promising tools for optimizing health status. They can provide valuable data to motivate and educate patients themselves, as well as for healthcare professionals, by analyzing health outcomes such as disease activity. Their uses raise various issues, such as patient beliefs about physical activity and connected devices, and their ability to interpret and analyze the data provided. Another challenge is the ability to analyze a large, continuous stream of data, which is now possible through the use of machine learning. We believe that the next few years will see many developments in this existing field, to the ultimate benefit of the patient.

## Figures and Tables

**Figure 1 sensors-20-04797-f001:**
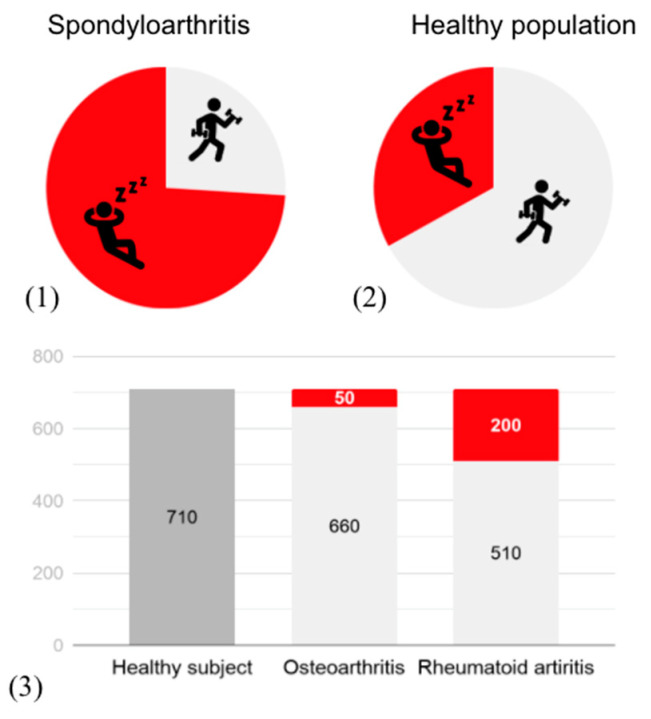
Physical activity level measured with questionnaire (**1**) (**2**) and with activity trackers (**3**). Y-axis = minutes of activity per week; the red part exposes the difference with the healthy population.

**Figure 2 sensors-20-04797-f002:**
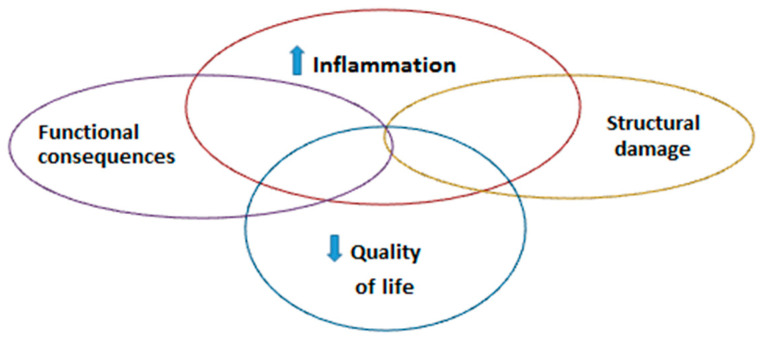
Consequences of flares.

**Figure 3 sensors-20-04797-f003:**
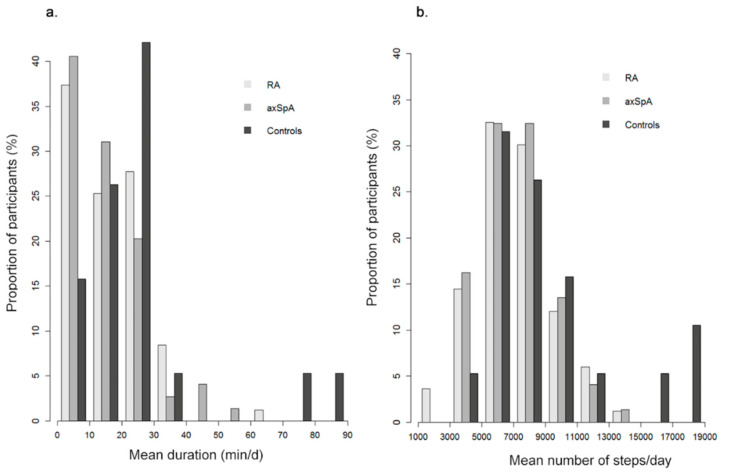
Distribution of mean physical activity over 3 months in the ActConnect Study. Distribution of mean physical activity over 3 months in 83 patients with rheumatoid arthritis, 74 with axial spondyloarthritis and 19 controls: (**a**) mean duration of moderate to vigorous activity (min/d) and (**b**) mean number of steps per day. RA: rheumatoid arthritis; axSpA: axial spondyloarthritis.

**Table 1 sensors-20-04797-t001:** Overview of rheumatoid arthritis and axial spondyloarthritis.

Rheumatoid Arthritis	Spondyloarthritis
Prevalence (ratio of the number of cases to the general population) 0.5% Usually occurs in females (70%), around the age of 50 years Cause: inflammation of the joints: arthritis Risk: progressive joint destruction Significant impact on quality of life: pain, functional disability, fatigue, but also psychosocial impact. Long-term immunosuppressive treatments (biologics) are often needed	Prevalence 0.3% Usually occurs in males (60%), around the age of 25 years Cause: inflammation of the spine Risk: possible damage to the spine (fusion of vertebrae, around 20%), eyes (uveitis), digestive tract (inflammatory bowel disease), skin (psoriasis) Significant impact on quality of life: pain, functional disability, fatigue Long-term immunosuppressive treatments (biologics) are less often needed

**Table 2 sensors-20-04797-t002:** Methods to assess disease flares in rheumatic diseases.

**Mode of Administration:**	–Physician report e.g., using single questions (e.g., is the patient flaring?) or based on imaging or lab results. –Patient report using single questions or standardized questionnaires (e.g., have you experienced a flare?) Or questionnaires such as the RA-Flare questionnaire [76].
**Focus on Increase of Inflammation:**	–Increase in symptoms and in particular night and morning pain. –Imaging findings (e.g., inflammation on magnetic resonance imaging of the spine in axSpA). –Physical examination e.g., increase in swollen joints in RA. –Increase of inflammation based on composite scores (which assess both symptoms and objective inflammation).
**Focus on Life Changes:**	–Change or optimization of treatment. –Medical consultation because of a flare. –Decrease of patient activity in daily life, changes in lifestyle.

RA, Rheumatoid Arthritis; axSpA, axial SpondyloArthritis.

**Table 3 sensors-20-04797-t003:** Prediction of flares by steps/hour using a Machine Learning approach on pooled analyses (corresponding to 4030 weeks of physical activity overall).

Number of Weeks	Flare According to the Patient (*N* = 920 Weeks)	No Patient-Reported Flare (*N* = 3110 Weeks)
Flare According to the Modelization by Machine-Learning	880	104
No Flare According to the Modelization	40	3006

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
