# Peer review of "Wearable Activity Trackers in the Management of Rheumatic Diseases: Where Are We in 2020?"

_sensors, 2020, doi:10.3390/s20174797_

Round 1

Reviewer 1 Report

I would like to thank the authors for presenting this review article, that summarizes use of wearable trackers in the management of rheumatic diseases. The paper presents the impact of wearable trackers on overall activity and health monitoring of patients and how they can be used to detect disease flares. Here are some minor comments and suggestions,

  1. In Table 1, define prevalence. 
  2. In Box 2, merge paragraphs, use single paragraph for each topic.
  3. Section 3.2 is not clear. Table 2 should be explained with more details, like what kind of questions were asked.
  4. There should be a separate subsection explaining how you formulated the machine learning problem.
  5. Explain the conversion of steps/hour to weeks.

Author Response

Response regarding the manuscript entitled, " Wearable activity trackers in the management of rheumatic diseases: where are we in 2020?" to Sensors

Paris, August 14, 2020

Dear Editors,

Thank you for giving us the opportunity to improve our paper. Please find below the point by point answers to the reviewers and attached our modified paper.

Best regards,

Thomas Davergne and Laure Gossec on behalf of the authors

Response to Reviewer 1 Comments

Point 1: Comments and Suggestions for Authors. I would like to thank the authors for presenting this review article, that summarizes use of wearable trackers in the management of rheumatic diseases. The paper presents the impact of wearable trackers on overall activity and health monitoring of patients and how they can be used to detect disease flares. Here are some minor comments and suggestions,

Response 1: we thank the Reviewer for this kind comment.

Point 2: In Table 1, define prevalence.

Response 2: We thank the Reviewer for pointing out this unclarity and have amended as proposed.

Action point 2: We added a short definition in table 1 p2: “Prevalence (ratio of the number of cases to the general population) 0.5%”.

Point 3: In Box 2, merge paragraphs, use single paragraph for each topic.

Response 3: We agree with this form comment and have modified as suggested.

Action Point 3: We merged paragraph in Box 2 p3, using a single paragraph for each topic:

For physical activity: At least 150 minutes of moderate-intensity endurance activity or at least 75 minutes of sustained-intensity endurance activity, or an equivalent combination of moderate and sustained-intensity activity, during the week. Endurance activity should be performed in periods of at least 10 minutes. For additional health benefits, adults should increase the duration of moderate-intensity endurance activity to 300 minutes per week or 150 minutes per week of sustained endurance activity, or an equivalent combination of moderate and sustained activity. Strengthening exercises involving the major muscle groups should be performed at least two days per week.

For sedentary lifestyles, it is recommended: To reduce the total daily time spent sitting or lying down (outside of sleeping and eating time) as much as possible. To walk for a few minutes and stretch after 2 hours in a row in a sitting or lying position and to make a few movements that activate the muscles and mobilize the joints (rotation of the shoulders, pelvis, ankles, wrists, hands, head).

Point 4: Section 3.2 is not clear. Table 2 should be explained with more details, like what kind of questions were asked.

Response 4: We thank the Reviewer for raising this unclarity. We adapted the paragraph to increase clarity. Details were added and the questions asked have been clarified in section 3.2 p7 and in the Table

Action point 4: Section 3.2 p7 and the Table are now as below:

Flares can be assessed by single questions or standardised questionnaires to detect increased inflammation or life changes.  (Table 2) [1,2]. For this assessment, the physician may ask questions directly to the patient during the clinical visit, or the patient may complete a questionnaire himself or herself without the physician's assistance.

Table 2 : Methods to assess disease flares in rheumatic diseases

Mode of administration:

- Physician report e.g. using single questions (e.g., is the patient flaring?) or based on imaging or lab results

- Patient report using single questions or standardised questionnaires (e.g., have you experienced a flare?) Or questionnaires such as the RA-Flare questionnaire [3]

Focus on increase of inflammation:

- Increase in symptoms and in particular night and morning pain

- Imaging findings (e.g., inflammation on magnetic resonance imaging of the spine in AxSpA)

- Physical examination e.g., increase in swollen joints in RA

- Increase of inflammation based on composite scores (which assess both symptoms and objective inflammation)

Focus on life changes:

- Change or optimization of treatment

- Medical consultation because of a flare

- Decrease of patient activity in daily life, changes in lifestyle

Point 5: There should be a separate subsection explaining how you formulated the machine learning problem.

Response 5: We understand the necessity of a separate subsection in this part of the paper and thank the Reviewer for this comment. We added the information as suggested

Action point 5: We added a subsection p9:

“Use of machine learning to enable more accurate analysis

In a second phase, we analysed the link between patient-reported flares and activity-tracker-provided steps per minute (and not mean steps per day) using artificial intelligence, and more specifically machine-learning through selective (multiclass) naive Bayesian statistical methods [4]. Machine learning allows analyses of huge amounts of data with minimal aggregation of data. Such techniques necessitate high computer power and dedicated programs. In this case, we worked with a telecommunications company, Orange Healthcare. “

Point 6: Explain the conversion of steps/hour to weeks.

Response 6: We thank the reviewer for this question which was an important methodological point. In the results of the modelling phase, the software of machine learning used (Khiops©) provides an evaluation of the importance (the weight) of each explanatory variable in the model. Using these weights, a time-line map of “significant” moments of activity during the week was created (data not shown). This time line shows the weight of the moments of the days used by the algorithm to perform the score calculation for the classification. The aggregation being hourly, these weights characterize the importance of each hour of each day of the week in the resulting flare classification.

Action point 6: We added this explanation p9:

“In the results of the modelling phase, the software of machine learning used (Khiops©) provides an evaluation of the importance (the weight) of each explanatory variable in the model. Using these weights, a time-line map of “significant” moments of activity during the week was created (data not shown). This time line shows the weight of the moments of the days used by the algorithm to perform the score calculation for the classification. The aggregation being hourly, these weights characterize the importance of each hour of each day of the week in the resulting flare classification.”

References:

1 van Gestel AM, Prevoo ML, van ’t Hof MA, et al. Development and validation of the European League Against Rheumatism response criteria for rheumatoid arthritis. Comparison with the preliminary American College of Rheumatology and the World Health Organization/International League Against Rheumatism Criteria. Arthritis Rheum 1996;39:34–40. doi:10.1002/art.1780390105

2 Molto A, Gossec L, Meghnathi B, et al. An Assessment in SpondyloArthritis International Society (ASAS)-endorsed definition of clinically important worsening in axial spondyloarthritis based on ASDAS. Ann Rheum Dis 2018;77:124–7. doi:10.1136/annrheumdis-2017-212178

3 Fautrel B, Morel J, Berthelot J-M, et al. Validation of FLARE-RA, a Self-Administered Tool to Detect Recent or Current Rheumatoid Arthritis Flare. Arthritis & Rheumatology (Hoboken, NJ) 2017;69:309–19. doi:10.1002/art.39850

4 Gossec L, Guyard F, Leroy D, et al. Detection of Flares by Decrease in Physical Activity, Collected Using Wearable Activity Trackers in Rheumatoid Arthritis or Axial Spondyloarthritis: An Application of Machine Learning Analyses in Rheumatology. Arthritis Care Res (Hoboken) 2019;71:1336–43. doi:10.1002/acr.23768

5 Mahloko L, Adebesin F. A Systematic Literature Review of the Factors that Influence the Accuracy of Consumer Wearable Health Device Data. In: Hattingh M, Matthee M, Smuts H, et al., eds. Responsible Design, Implementation and Use of Information and Communication Technology. Cham: : Springer International Publishing 2020. 96–107. doi:10.1007/978-3-030-45002-1_9

6 Cosoli G, Spinsante S, Scalise L. Wrist-worn and chest-strap wearable devices: Systematic review on accuracy and metrological characteristics. Measurement 2020;159:107789. doi:10.1016/j.measurement.2020.107789

7 Shin G, Jarrahi MH, Fei Y, et al. Wearable activity trackers, accuracy, adoption, acceptance and health impact: A systematic literature review. Journal of Biomedical Informatics 2019;93:103153. doi:10.1016/j.jbi.2019.103153

8 Chu AHY, Ng SHX, Paknezhad M, et al. Comparison of wrist-worn Fitbit Flex and waist-worn ActiGraph for measuring steps in free-living adults. PLOS ONE 2017;12:e0172535. doi:10.1371/journal.pone.0172535

9 Chow H-W, Yang C-C. Accuracy of Optical Heart Rate Sensing Technology in Wearable Fitness Trackers for Young and Older Adults: Validation and Comparison Study. JMIR mHealth and uHealth 2020;8:e14707. doi:10.2196/14707

Reviewer 2 Report

Overall this manuscript has the potential to publication if the minor issues should be addressed.

- The author may consider research design as a 'review' rather than the 'perspective' study.

- This study seems to introduce the authors' previous studies (ActConnect study). Was there any difference between self-monitoring and external monitoring by health professionals in the physical activity monitoring and health benefit of rheumatic patients?

- Line 71-74: Although this manuscript is clear and well-written, there is insufficient information in the objective section.

- Line 83-118: This paragraph is a general guideline for physical activity to promote the health of healthy adults. I am wondering if this paragraph is necessary. It is necessary to compare the physical activity of a person with rheumatic disease and the healthy participants, but I believe the author should focus more on the health benefits of using wearable activity trackers in patient with rheumatic disease.

- Minor revision

Line 44: Please add the references.

Line 71-74: Please change to the past tense.

Author Response

Response regarding the manuscript entitled, " Wearable activity trackers in the management of rheumatic diseases: where are we in 2020?" to Sensors

Paris, August 14, 2020

Dear Editors,

Thank you for giving us the opportunity to improve our paper. Please find below the point by point answers to the reviewers and attached our modified paper.

Best regards,

Thomas Davergne and Laure Gossec on behalf of the authors

Response to Reviewer 2 Comments

Point 1: Comments and Suggestions for Authors. Overall this manuscript has the potential to publication if the minor issues should be addressed.

Response 1: we thank the Reviewer for this kind comment.

Point 2: The author may consider research design as a 'review' rather than the 'perspective' study.

Response 2: we thank the Reviewer for this point. We would like to note that the editors asked us to write a ‘perspective paper’ therefore the editors should decide how to call this paper.

Action point 2: we changed the research design as a “review” rather than a “perspective” but the editors must validate this.

Point 3: This study seems to introduce the authors' previous studies (ActConnect study). Was there any difference between self-monitoring and external monitoring by health professionals in the physical activity monitoring and health benefit of rheumatic patients?

Response 3: We thank the Reviewer for raising this point. In this review, we presented self-monitoring and external monitoring as two different concepts, using the same tools (the activity tracker) but leading to different objectives. In self-monitoring, data collected by activity tracker are presented directly to the patient in order to improve a certain health behavior, here the physical activity level. In external-monitoring, data are used by the physician to assess the patient condition, in this case presence or absence of flare.

Point 4: Line 71-74: Although this manuscript is clear and well-written, there is insufficient information in the objective section:

Response 4: We thank the Reviewer for this observation on which we agree on.

Action point 4: based on the comment, we added information in the objective session p2:

“Objectives

In this review, we will discuss the roles of connected activity trackers for the management of patients with rheumatic and musculoskeletal disorders, in 2020. First, we will discuss the use of activity trackers for self-monitoring to promote physical activity in patients at risk of inactivity. The need to consider behaviour change techniques and address barriers and facilitators, in addition to the use of activity trackers, will be explained. We will then discuss the use of activity trackers as a promising tool for detecting flares in patients with inflammatory arthritis. We will introduce the use of machine learning to enable more accurate analysis of data to detect flares. Although the promising use of activity trackers for self-monitoring and external monitoring, various limits exists and will be address at the end of this review.”

Point 5: Line 83-118: This paragraph is a general guideline for physical activity to promote the health of healthy adults. I am wondering if this paragraph is necessary. It is necessary to compare the physical activity of a person with rheumatic disease and the healthy participants, but I believe the author should focus more on the health benefits of using wearable activity trackers in patient with rheumatic disease.

Response 5: We thank the reviewer for raising this point. Since this review deals with physical activity behaviour and strategies to increase physical activity levels, we want to ensure that the reader has a clear definition of what physical activity is and what inactivity is. In addition, in order to correctly interpret the effect of activity monitoring on physical activity levels, we want to outline the current physical activity recommendations to be achieved. Therefore we respectfully have not deleted this part.

Point 6: Minor revision. Line 44: Please add the references.

Response 6: We thank the reviewer for this comment. We have added a general reference as asked.

Action point 6: we added a reference: European League Against Rheumatism (EULAR) Taskforce 2017. RheumaMap: a Research Roadmap to transform the lives of people with Rheumatic and Musculoskeletal Diseases.

Point 7: Line 71-74: Please change to the past tense.

Response 7: We thank the Reviewer for this comment. However as this paragraph is announcing the plan and objectives of the paper, it would be strange to write this as past tense. We have therefore modified from the future to the present tense and hope this is what the reviewer wanted.

Action point 7: We changed the paragraph p2 into: “In this review, we discuss the roles of connected activity trackers for the management of patients with rheumatic and musculoskeletal disorders, in 2020. First, we discuss the use of activity trackers for self-monitoring to promote physical activity in patients at risk of inactivity. The need to consider behaviour change techniques and address barriers and facilitators, in addition to the use of activity trackers, is explained. We then discuss the use of activity trackers as a promising tool for detecting flares in patients with inflammatory arthritis. We introduce the use of machine learning to enable more accurate analysis of data to detect flares. Although the promising use of activity tracker for self-monitoring and external monitoring, various limits exists and are addressed at the end of this review.”

Reviewer 3 Report

Manuscript ID: sensors-892205

Title: Wearable activity trackers in the management of rheumatic diseases: where are we in 2020?

Recommendation: Minor revision

Brief summary

The topic of this manuscript is the monitoring of physical activity by means of wearable devices in the context of rheumatic disease. Wearables are able to educate and motivate patients, helping the clinician in their management. In particular, the authors employed machine learning techniques to demonstrate that physical activity is strongly related to disease flares. Many interesting considerations on the use of wearables are drafted in a dedicated section.

Broad comments

The topic is relevant, since wearable devices are widely used nowadays, also in clinical applications, revealing useful for the patients management.

The presentation is clear, the paper is well-written and the English is fluent, even if re-reading the paper could help to correct some typos and improve the use of punctuation somewhere.

The article is quite well contextualized in the literature background, even if some references could be added, in particular related to metrological aspects of sensors - also in the field of Sensors.

Some suggestions are provided in the next comments, which may help the authors in improving the quality of this paper.

Specific comments

Abstract: some quantitative data related to the results of the pilot study should be reported to give more impact to the article presentation. Maybe less space should be dedicated to the state of the art, emphasizing a little more the aspects related to wearable devices performance form a metrological point of view. Some examples of these devices limitations could be mentioned, as well.

Introduction: at the end of this section, it could be useful to briefly explain the organisation of the article sections, to improve the document readability. Moreover, some data about the prevalence of rheumatic disease should be reported within the text (in addition to Table 1) to better contextualize the paper.

Line 53: this line spacing should be eliminated, maybe adding one before the table caption. The same for Table 2 (line 273).

Table 1: I would eliminate the full stop at the end of the table.

Line 75: maybe the title of this paragraph could be made more explicative, referencing to physical activity monitoring instead of general self-monitoring.

Line 83: please remove the spacing before the colon.

Lines 85-86: a reference should be added.

Lines 132-133: please homogenise the title with the previous ones (bold for the first words, in particular).

Figure 1: I would place the figure centred. However, it is only a matter of style.

Paragraph 2/Paragraph2.4: something about the accuracy of wearable activity trackers should be reported, since it undoubtedly influences the results of the monitoring and the reliability of its interpretation. The authors could refer to some of the latest review on this topic, such as the ones reported hereafter:

  • Cosoli, G., Spinsante, S., & Scalise, L. (2020). Wrist-worn and chest-strap wearable devices: Systematic review on accuracy and metrological characteristics. Measurement, 107789.
  • Chow, H. W., & Yang, C. C. (2020). Accuracy of Optical Heart Rate Sensing Technology in Wearable Fitness Trackers for Young and Older Adults: Validation and Comparison Study. JMIR mHealth and uHealth, 8(4), e14707.
  • Mahloko, L., & Adebesin, F. (2020, April). A Systematic Literature Review of the Factors that Influence the Accuracy of Consumer Wearable Health Device Data. In Conference on e-Business, e-Services and e-Society (pp. 96-107). Springer, Cham.

Line 203: it could be useful to report the models of the wearable devices employed in the considered studies. Moreover, something about the study protocols adopted should be said (e.g. are they the same for all the studies? Which could be the possible influence on results?).

Line 204: a brief definition of “adherence” should be reported.

Line 232: please check the correctness of the verbs used in the sentence.

Lines 255-257: please check the correctness of the figures numbering.

Table 2: please check the formatting of the table, since it may result a bit difficult to read.

Line 286: this statement could be reported above, when speaking about activity tracker capability (lines 165-166), and briefly recalled here to introduce the performed study.

Line 295: the number and the main characteristics (e.g. gender and age) of the patients involved in the study should be reported.

Lines 314-318: please check the correctness of the figures numbering.

Figure 2: please improve the image resolution, if possible. Moreover, it would be better to put all the figure information in its caption, instead of some pieces of information below (lines 321-323), for the sake of readability.

Line 327-328: a brief definition and related reference should be added in relation to “nlme package”. Moreover, it could be better to put “nlme” in Italics, for a better readability.

Lines 346-348: brief definitions for the performance indices (e.g. sensitivity and specificity) should be reported, or just a reference.

Line 353: maybe this paragraph could be numbered 3.4.1, since it refers to what reported in 3.4.

Paragraph 4.1: some considerations on the accuracy of wearable devices (maybe considering the above-mentioned review paper on the topic) and its effect on results should be added. Other interesting literature papers that the authors could consider are the following ones:

  • Ameen, M.S.; Cheung, L.M.; Hauser, T.; Hahn, M.A.; Schabus, M. About the Accuracy and Problems of Consumer Devices in the Assessment of Sleep. Sensors 2019, 19, 4160.
  • Leth, S.; Hansen, J.; Nielsen, O.W.; Dinesen, B. Evaluation of Commercial Self-Monitoring Devices for Clinical Purposes: Results from the Future Patient Trial, Phase I. Sensors 2017, 17, 211.
  • Dijkhuis, T.B.; Blaauw, F.J.; Van Ittersum, M.W.; Velthuijsen, H.; Aiello, M. Personalized Physical Activity Coaching: A Machine Learning Approach. Sensors 2018, 18, 623.

Line 415: a line spacing should be added, for homogeneity with other paragraphs.

Line 431: please remove the space before the full stop.

Line 456: please remove the space before the colon.

Author Response

Response regarding the manuscript entitled, " Wearable activity trackers in the management of rheumatic diseases: where are we in 2020?" to Sensors

Paris, August 14, 2020

Dear Editors,

Thank you for giving us the opportunity to improve our paper. Please find below the point by point answers to the reviewers and attached our modified paper.

Best regards,

Thomas Davergne and Laure Gossec on behalf of the authors

Response to Reviewer 3 Comments

Point 1: Brief summary: The topic of this manuscript is the monitoring of physical activity by means of wearable devices in the context of rheumatic disease. Wearables are able to educate and motivate patients, helping the clinician in their management. In particular, the authors employed machine learning techniques to demonstrate that physical activity is strongly related to disease flares. Many interesting considerations on the use of wearables are drafted in a dedicated section.

Broad comments: The topic is relevant, since wearable devices are widely used nowadays, also in clinical applications, revealing useful for the patients management. The presentation is clear, the paper is well-written and the English is fluent, even if re-reading the paper could help to correct some typos and improve the use of punctuation somewhere. The article is quite well contextualized in the literature background, even if some references could be added, in particular related to metrological aspects of sensors - also in the field of Sensors. Some suggestions are provided in the next comments, which may help the authors in improving the quality of this paper.

Response 1: we thank the Reviewer for this kind comment and full review, and for giving us the opportunity to improve our paper.

Specific comments

Point 2: Abstract: some quantitative data related to the results of the pilot study should be reported to give more impact to the article presentation. Maybe less space should be dedicated to the state of the art, emphasizing a little more the aspects related to wearable devices performance form a metrological point of view. Some examples of these devices limitations could be mentioned, as well.

Response 2: We thank the Reviewer for this comment. We changed the abstract in p1 based on the comments, as proposed

Action point 2 : The abstract now states:

“In health care, physical activity can be monitored in two ways: self-monitoring by the patient himself or external monitoring by health professionals. Regarding self-monitoring, wearable activity trackers allow automated passive data collection that educate and motivate patients. Wearing an activity tracker can improve walking time by around 1500 steps per day. However, there are concerns about measurement accuracy (e.g. lack of a common validation protocol or measurement discrepancies between different devices). For external monitoring, many innovative electronic tools are currently used in rheumatology to help to support physician time management, to reduce the burden on clinic time, and to prioritize patients who may need further attention. In inflammatory arthritis, such as rheumatoid arthritis, regular monitoring of patients to detect disease flares improves outcomes. In a pilot study applying machine learning to activity tracker steps, we showed that physical activity was strongly linked to disease flares and that patterns of physical activity could be used to predict flares with great accuracy, with a sensitivity and specificity above 95%. Thus, automatic monitoring of steps may lead to improved disease control through potential early identification of disease flares. However, activity trackers have some limitations when applied to rheumatic patients, such as tracker adherence, lack of clarity on long-term effectiveness or the potential multiplicity of trackers.”

Point 3: Introduction: at the end of this section, it could be useful to briefly explain the organisation of the article sections, to improve the document readability. Moreover, some data about the prevalence of rheumatic disease should be reported within the text (in addition to Table 1) to better contextualize the paper.

Response 3: We thank the Reviewer for this observation on which we agree on.

Action point 4: based on the comment, we added information at the end of introduction in the objective session p2:

“Objectives

In this review, we discuss the roles of connected activity trackers for the management of patients with rheumatic and musculoskeletal disorders, in 2020. First, we discuss the use of activity trackers for self-monitoring to promote physical activity in patients at risk of inactivity. The need to consider behaviour change techniques and address barriers and facilitators, in addition to the use of activity trackers, is explained. We then discuss the use of activity trackers as a promising tool for detecting flares in patients with inflammatory arthritis. We introduce the use of machine learning to enable more accurate analysis of data to detect flares. Although the promising use of activity tracker for self-monitoring and external monitoring, various limits exists and are addressed at the end of this review.”

We also added in the text p2: “The prevalence of these two conditions is 0.5% and 0.3% respectively.”

Point 4: Line 53: this line spacing should be eliminated, maybe adding one before the table caption. The same for Table 2 (line 273).

Response 4: We agree on this observation and we thank the Reviewer for this comment.

Action point 4: line spacing were eliminated line 53 and 273.

Point 5: Table 1: I would eliminate the full stop at the end of the table.

Response 5: We thank the reviewer for this observation.

Action point 5: We eliminated the full stop at the end of the table 1.

Point 6: Line 75: maybe the title of this paragraph could be made more explicative, referencing to physical activity monitoring instead of general self-monitoring.

Response 6: We understand the reviewer’s point.

Action point 6: we changed the title p2 into: “Use of activity trackers for self-monitoring of physical activity”.

Point 7: Line 83: please remove the spacing before the colon.

Response 7: We thank the Reviewer for this observation.

Action point 7: We removed the space before the colon L 83.

Point 8: Lines 85-86: a reference should be added.

Response 8: We thank the Reviewer for this observation.

Action point 8: We added a reference line 85-86 “Caspersen, C.J.; Powell, K.E.; Christenson, G.M. Physical activity, exercise, and physical fitness: definitions and distinctions for health-related research. Public Health Rep 1985, 100, 126–131”

Point 9: Lines 132-133: please homogenise the title with the previous ones (bold for the first words, in particular).

Response 9: We thank the Reviewer for this comment.

Action point 9: We changed the title of the Box 3 into: “Box 3: Overview of benefits of physical activity in the general population and in patients with rheumatic and musculoskeletal diseases.”

Point 10: Figure 1: I would place the figure centred. However, it is only a matter of style.

Response 10: We thank the Reviewer for this observation.

Action point 10: We centred the figure 1 as recommended.

Point 11: Paragraph 2/Paragraph2.4: something about the accuracy of wearable activity trackers should be reported, since it undoubtedly influences the results of the monitoring and the reliability of its interpretation. The authors could refer to some of the latest review on this topic, such as the ones reported hereafter:

Cosoli, G., Spinsante, S., & Scalise, L. (2020). Wrist-worn and chest-strap wearable devices: Systematic review on accuracy and metrological characteristics. Measurement, 107789.

Chow, H. W., & Yang, C. C. (2020). Accuracy of Optical Heart Rate Sensing Technology in Wearable Fitness Trackers for Young and Older Adults: Validation and Comparison Study. JMIR mHealth and uHealth, 8(4), e14707.

Mahloko, L., & Adebesin, F. (2020, April). A Systematic Literature Review of the Factors that Influence the Accuracy of Consumer Wearable Health Device Data. In Conference on e-Business, e-Services and e-Society (pp. 96-107). Springer, Cham.

Response 11: This is an important methodological point and we thank the reviewer for having raised it.

Action point 11: We added a paragraph on this topic p5

“2.5. Accuracy of activity trackers

Although wearable activity trackers have seen an upward trend worldwide, there are concerns over the accuracy of the data collected by these devices. Factors that influence the accuracy of the data collected can be classified into 3 categories, namely (I) the tracker and sensor type, (II) the algorithm used in the device, and (III) the limitation in the design, energy consumption, and processing capability of the device [1]. Another issue with accuracy of activity trackers is that there is no consensus on standard procedures to validate accuracy of activity tracker [2]. A common approach is the comparison of multiple devices, or across devices or mobile applications, in terms of the accuracy [3]. This comparison is usually conducted in laboratory settings, with a focus on physical activity such as walking on a treadmill or normal jogging, rather than investigating the technology in natural settings [3]. However, one study including 104 participants conducted an experience in free-living conditions during 7 days. The average number of steps per day was compared between the wrist-worn Fitbit Flex and waist-worn ActiGraph (wGT3X-BT) considered here as a reference standard [4]. This study showed a high correlation and agreement in steps between Fitbit Flex and ActiGraph, while discrepancies were observed between devices in average steps/day (Fitbit Flex, 10193; ActiGraph, 8812).”

Point 12: Line 204: a brief definition of “adherence” should be reported.

Response 12: We thank the Reviewer for this observation and have modified as suggested.

Action point 12: We added Line 204 a shot definition of adherence used in the related study: “The results showed that short-term adherence to wearable activity trackers was high in this population. Here, adherence was considered as ( duration of use and percentage of patients still in the study at completion [study completion rate]).”

Point 13: Line 232: please check the correctness of the verbs used in the sentence.

Response 13: We thank the Reviewer for raising this mistake.

Action point 13: We corrected the sentence Line 232 into : “These elements are important to consider in a comprehensive approach to improve physical activity and should be associated with the use of activity trackers.”

Point 14: Lines 255-257: please check the correctness of the figures numbering.

Response 14: We thank the Reviewer for pointing out this error.

Action point 14: We changed the figure numbering into “Figure 2”

Point 15: Table 2: please check the formatting of the table, since it may result a bit difficult to read.

Response 15: We thank the Reviewer for this observation. We have changed Table 2 to a more user-friendly format and modifie dht econtent as suggested by another reviewer.

Action point 15::Table 2 now reads:

Mode of administration:

- Physician report e.g. using single questions (e.g., is the patient flaring?) or based on imaging or lab results

- Patient report using single questions or standardised questionnaires (e.g., have you experienced a flare? Or questionnaires such as the RA-Flare questionnaire [5]

Focus on increase of inflammation:

- Increase in symptoms and in particular night and morning pain

- Imaging findings (e.g., inflammation on magnetic resonance imaging of the spine in AxSpA)

- Physical examination e.g., increase in swollen joints in RA

- Increase of inflammation based on composite scores (which assess both symptoms and objective inflammation)

Focus on life changes:

- Change or optimization of treatment

- Medical consultation because of a flare

- Decrease of patient activity in daily life, changes in lifestyle

Point 16: Line 286: this statement could be reported above, when speaking about activity tracker capability (lines 165-166), and briefly recalled here to introduce the performed study.

Response 16: We thank the Reviewer for this comment on which we agree on. As explained above, a paragraph has been added in section 2.5 related to accuracy of activity tracker.

Action point 16: Based on this comment, we changed Line 286 the sentence to : “Physical activity can be objectively measured using activity trackers, although there is some uncertainty about the accuracy of measurement (see section 2.5).”

Point 17: Line 295: the number and the main characteristics (e.g. gender and age) of the patients involved in the study should be reported.

Response 17: We thank the Reviewer for this observation.

Action point 17: We added the following sentence Line 292 : “A total of 157 patients (83 rheumatoid arthritis and 74 axial spondyloarthritis) were analysed; 36.3% (57/157) patients were males, and their mean age was 46 (standard deviation [SD] 12) years and mean disease duration was 11 (SD 9) years.”

Point 18: Lines 314-318: please check the correctness of the figures numbering.

Response 18: We thank the Reviewer for pointing out this error.

Action point 18: We changed the figure numbering into “Figure 3”

Point 19: Figure 2: please improve the image resolution, if possible. Moreover, it would be better to put all the figure information in its caption, instead of some pieces of information below (lines 321-323), for the sake of readability.

Response 19: We thank the Reviewer for this observation.

Action point 19: We increased the image resolution as requested and put all the figure information in the caption p10.

Point 20: Line 327-328: a brief definition and related reference should be added in relation to “nlme package”. Moreover, it could be better to put “nlme” in Italics, for a better readability.

Response 20: We thank the Reviewer for this comment on which we agree on.

Action point 20: We added the following sentence p10: “The nlme package (for Linear and Nonlinear Mixed Effects Models) allows to fit and compare Gaussian linear and nonlinear mixed-effects models [1]”.

Point 21: Lines 346-348: brief definitions for the performance indices (e.g. sensitivity and specificity) should be reported, or just a reference.

Response 21: We thank the Reviewer for this observation.

Action point 21: We added the following information p11: “It is remarkable that the machine learning model detected correctly both patient-reported flares and absence of flares with a sensitivity (the ability of a test to correctly classify an individual as flaring) of 96% and a specificity (the ability of a test to correctly classify an individual as not flaring) of 97% [1].”

Point 22: Line 353: maybe this paragraph could be numbered 3.4.1, since it refers to what reported in 3.4.

Response 22: We understand the reviewer’s point.

Action point 22: We changed the format of this section p11 to include it in 3.4 as proposed.

Point 23: Paragraph 4.1: some considerations on the accuracy of wearable devices (maybe considering the above-mentioned review paper on the topic) and its effect on results should be added. Other interesting literature papers that the authors could consider are the following ones:

Ameen, M.S.; Cheung, L.M.; Hauser, T.; Hahn, M.A.; Schabus, M. About the Accuracy and Problems of Consumer Devices in the Assessment of Sleep. Sensors 2019, 19, 4160.

Leth, S.; Hansen, J.; Nielsen, O.W.; Dinesen, B. Evaluation of Commercial Self-Monitoring Devices for Clinical Purposes: Results from the Future Patient Trial, Phase I. Sensors 2017, 17, 211.

Dijkhuis, T.B.; Blaauw, F.J.; Van Ittersum, M.W.; Velthuijsen, H.; Aiello, M. Personalized Physical Activity Coaching: A Machine Learning Approach. Sensors 2018, 18, 623.

Line 415: a line spacing should be added, for homogeneity with other paragraphs.

Response 23: We thank the Reviewer for this insightful comment on which we do agree on and have added this point to the paragraph 4.1.

Action point 23: We added the following paragraph in p13: “Accuracy of activity trackers. As mentioned above in point 2.5, the accuracy of activity tracker remains a concern and must be taken into account when analysing results. The accuracy of step counting can be influenced by the walking speed. In a study comparing the number of steps measured by commercial activity trackers versus gyroscopic readings at 2 km/h, the highest error showed a margin of 26.8% with the Fitbit HR load, while at 3.5 km/h, the highest error showed a margin of 1.5% with the Fitbit One [1].”

Point 24: Line 431: please remove the space before the full stop.

Response 24: We thank the Reviewer for this observation.

Action point 24: We removed the space before the full stop Line 431.

Point 25: Line 456: please remove the space before the colon.

Response 25: We thank the Reviewer for this observation.

Action point 25: We removed the space before the colon L 456.

Point 26: Line 203: it could be useful to report the models of the wearable devices employed in the considered studies. Moreover, something about the study protocols adopted should be said (e.g. are they the same for all the studies? Which could be the possible influence on results?).

Response 26: We thank the Reviewer for raising this point.

Action point 26: We reported models of the wearable devices employed in the considered studies and some considerations about protocols adopted Line 203 :  “Overall, 9 studies used a pedometer (KenzLifecoder EX, Suzuken Co, Yamax Digiwalker Pedometer Model SW-200 (x2), Yamax Digiwalker CW-701, Omron HJ720ITC (x2), Mt-x movement sensor and a PDA or non-specified) and 8 a more advanced activity tracker (Jawbone UP 24, Fitbit (Fitbit Flex, Fitbit Charge HR, Fitbit Zip (x2) or non-specified (x3)). All studies but 1 used co-interventions in addition to the use of activity tracker. Among them, 12 studies (71%) used goal setting, 9 (53%) used educational walking booklets, and 8 (47%) used counselling. The weighted mean duration of intervention was 21.8 weeks (range 2.0–52.0). Activity trackers were worn on the wrist (6 studies [35%]) and waist (11 studies [65%]). Of 13 studies measuring steps per day, the measurement was performed by the WAT itself in 9 studies (69%), by a research device (e.g., Actigraph) in 3 studies (23%), and by both in 1 study (8%).”

Point 27: Line 415: a line spacing should be added, for homogeneity with other paragraphs.

Response 27: We thank the Reviewer for this observation.

Action point 27:  We added a line spacing Line 415.

References:

1          Mahloko L, Adebesin F. A Systematic Literature Review of the Factors that Influence the Accuracy of Consumer Wearable Health Device Data. In: Hattingh M, Matthee M, Smuts H, et al., eds. Responsible Design, Implementation and Use of Information and Communication Technology. Cham: : Springer International Publishing 2020. 96–107. doi:10.1007/978-3-030-45002-1_9

2          Cosoli G, Spinsante S, Scalise L. Wrist-worn and chest-strap wearable devices: Systematic review on accuracy and metrological characteristics. Measurement 2020;159:107789. doi:10.1016/j.measurement.2020.107789

3          Shin G, Jarrahi MH, Fei Y, et al. Wearable activity trackers, accuracy, adoption, acceptance and health impact: A systematic literature review. Journal of Biomedical Informatics 2019;93:103153. doi:10.1016/j.jbi.2019.103153

4          Chu AHY, Ng SHX, Paknezhad M, et al. Comparison of wrist-worn Fitbit Flex and waist-worn ActiGraph for measuring steps in free-living adults. PLOS ONE 2017;12:e0172535. doi:10.1371/journal.pone.0172535

5          Fautrel B, Morel J, Berthelot J-M, et al. Validation of FLARE-RA, a Self-Administered Tool to Detect Recent or Current Rheumatoid Arthritis Flare. Arthritis & Rheumatology (Hoboken, NJ) 2017;69:309–19. doi:10.1002/art.39850
